# Life Cycle Assessment of Dietary Patterns in the United States: A Full Food Supply Chain Perspective

**Daesoo Kim**, **Ranjan Parajuli** and **Gregory J. Thoma ***

Ralph E. Martin Department of Chemical Engineering, University of Arkansas, Fayetteville, AR 72701, USA;
dskim@uark.edu (D.K.); rparajul@uark.edu (R.P.)
***** Correspondence: gthoma@uark.edu; Tel.: +1-479-575-4951

**Abstract:** A tiered hybrid input–output-based life cycle assessment (LCA) was conducted to analyze potential environmental impacts associated with current US food consumption patterns and the recommended USDA food consumption patterns. The greenhouse gas emissions (GHGEs) in the current consumption pattern (CFP 2547 kcal) and the USDA recommended food consumption pattern (RFP 2000 kcal) were 8.80 and 9.61 tons $CO_2$-eq per household per year, respectively. Unlike adopting a vegetarian diet (i.e., RFP 2000 kcal veg or RFP 2600 kcal veg), adoption of a RFP 2000 kcal diet has a probability of increasing GHGEs and other environmental impacts under iso-caloric analysis. The bigger environmental impacts of non-vegetarian RFP scenarios were largely attributable to supply chain activities and food losses at retail and consumer levels. However, the RFP 2000 vegetarian diet showed a significant reduction in the environmental impacts (e.g., GHGEs were 22% lower than CFP 2547). Uncertainty analysis confirmed that the RFP 2600 scenario (mean of 11.2; range 10.3–12.4 tons $CO_2$-eq per household per year) is higher than CFP 2547 (mean of 8.81; range 7.89–9.95 tons $CO_2$-eq per household per year) with 95% confidence. The outcomes highlight the importance of incorporating environmental sustainability into dietary guidelines through the entire life cycle of the food system with a full accounting of the effects of food loss/waste.

**Keywords:** life cycle assessment; environmental impact; greenhouse gas emission; input-output; food supply chain; dietary guideline

---

## 1. Introduction

Food security and sustainability has been described as an emerging challenge by researchers, policy makers, producers, manufacturing companies, retailers, and consumers. Sustainability is complex and requires a multidisciplinary approach to account for the various factors involved. The general principle of sustainability is that we should consume resources at rates that do not exceed the capacity of Earth to replace these resources [1]. A prevailing theme in agroecological/food systems is maximizing food production efficiency (sustainable intensification) and availability through minimizing waste and losses of valuable commodities. The global population is projected to grow both in numbers and wealth in the coming decades. Over the same period, food production faces challenges from climate change, competing land uses, and shrinking water supplies. Thus, meeting food demand in a sustainable manner requires that we greatly increase the amount of food produced with limited additional resource inputs and improve supply chain efficiency in order to reduce food loss and waste. However, according to the UN FAO reports, a third of total food produced for human consumption, about 1.3 billion tons per year, is lost globally [2]. Cuéllar and Webber (2010) reported that approximately 27% of edible food in the United States was wasted [3]. Bernstad Saraiva Schott and Andersson (2015) indicated that about 35% of household food waste in Sweden is avoidable [4]. Hall et al. (2009) estimated food loss using a mathematical model of body weight and metabolism, and concluded that up to 40% of

all the food produced in the US goes uneaten [5]. Buzby and Hyman (2012) assessed the total value of food loss to be equivalent to $166 billion each year in the US, and this lost food ends up decaying in landfills as one of the largest fractions of municipal solid waste [6]. Nevertheless, disagreement exists between researchers regarding the quantification of food loss, as substantial quantities of food are lost or wasted throughout the entire life cycle of the supply chain, from agricultural production to final household consumption. A significant amount of the resources used in food production, processing, transport, storage, and consumption is wasted, resulting in avoidable economic and environmental impacts. Therefore, a better understanding of the rate and degree of food production, consumption, and loss in each supply chain stage as a function of dietary patterns, as well as the life cycle assessment (LCA) of food systems, will help us to understand the avenues that delineate opportunities to maximize the utilization of our resources, reduce environmental impacts, and support sustainably responsible commitments.

A number of LCA studies and review articles related to food supply chains have been reported worldwide [7–19]. There are some deficiencies in detailed environmental impact assessment across the entire food system and these are variously evaluated with different methodological approaches. For instance, many studies only focus on the analysis of greenhouse gas emissions (GHGEs), system delimitations are varied, and choices of metrics for the evaluations are also different. Numerous studies take the system boundary only up to the farm gate or retail stage into account because the production phase generally has the largest environmental impact [20–23]. However, for foods that have small environmental impacts during production, ignoring activities after farm gate may have a sizeable effect [24]. Although several studies concluded that adopting the dietary guidelines would diminish the overall environmental impacts [18,25,26], there is not unanimous agreement [11,14,19]. There exist research gaps and dissimilarities of current food consumption patterns (CFP) and recommended food consumption patterns (RFP) among different countries. Gruber et al. (2016) underlined the significance of including the consumer stage in food-related LCAs [27]. It is widely argued that animal products have higher impacts compared to plant-based products, however, there is evidence that there exist significant differences among animal products [28,29]. The differences in the impacts are further complicated when they are compared based on their nutritional contents [30–32]. Vieux et al. (2013) concluded that, despite a self-selected diet with high nutritional quality containing large amount of plant-based food proteins, they did not end up with the lowest GHG emissions [26]. Therefore, this study expands the boundary conditions and presents comprehensive environmental impact profiles in a cradle-to-grave perspective for the food supply chain in the United States. The evaluation has thus followed the steps of agricultural production through consumption with in-depth inventory analysis of retail and consumer phases, which were built in addition to the primary production (farm to processing).

The overarching objective of the study is to quantify differences in the potential environmental impacts of the food supply chain systems following the CFP and the RFPs. The first task of the study was thus to assess the total food supply chain, including the production, consumption and associated loss, by commodity class. The losses were calculated based on the proportion of food losses in the CFP across the supply chain. The results are presented for the recommended food consumption patterns, assuming a situation of a typical household in the United States following the USDA dietary recommendations [33].

## 2. Methods

The main approach is based on a tiered hybrid LCA of the food system constructed from a national environmentally-extended input–output model (EIO-LCA) up to retail gate, and on a process-based LCA model for the retail and consumer phases [34–36]. The comprehensive environmental data archive (CEDA) was used for the assessment [37]. The input–output model represents nationwide financial transitions between industrial sectors, hence it is appropriate to account for the balance sheet at national level [15]. Hendrie et al. (2014) point out that process-based LCA lacks upstream process completeness

due to the application of a system boundary that necessarily excludes some (small) processes [15]. The CEDA model does not include international trade flows, and this is a limitation for this study; most food consumed in the US is produced in the US, and imported food is modeled as domestic production. As Jones et al. (2008) addressed, EIO-LCA lacks coverage beyond the manufacturing stage [16]. In addition, the environmental burden related to the retail stage derived from input–output (I/O) tables cannot be allocated to specific food groups, thus this study adopted a process-based LCA model for retail and consumer level analysis to improve the resolution in combination with EIO tables. The potential environmental impacts of each individual food group reported in the US Department of Agriculture (USDA) Economic Research Service (ERS) loss-adjusted food availability (LAFA) database were analyzed by mapping food availability and losses across each life cycle stage [38,39]. Figure 1 is a schematic of the food supply chain system including the percentage of food loss, summed across all food groups, in each stage on a mass basis. The life cycle inventory (LCI) evaluation followed the USDA definitions [40] to account the commodity flow and product loss, such as primary (production and processing); retail; consumer transport and storage of products prior to consumption; and landfilling of food waste and packaging materials after disposal. Specific tasks required for this effort included estimation of the economic value (purchase price) for each food group in the typical US household diet as well as an estimate of the quantity consumed. There were other publicly available estimates of food consumption patterns, most notably from the National Cancer Institute and the US Census Bureau [41,42]. These estimates of food consumption were evaluated as part of this study; however, it was found that these data were not consistent with the reported loss rates and primary agricultural production data available from the USDA ERS. Specifically, using available loss rates and reported consumption to estimate the required production did not reproduce the reported primary production, thus these datasets are not internally consistent across the supply chain. In addition, these two sources of consumption data tended to bracket the ERS dataset, therefore this analysis was based strictly on the USDA ERS LAFA database.

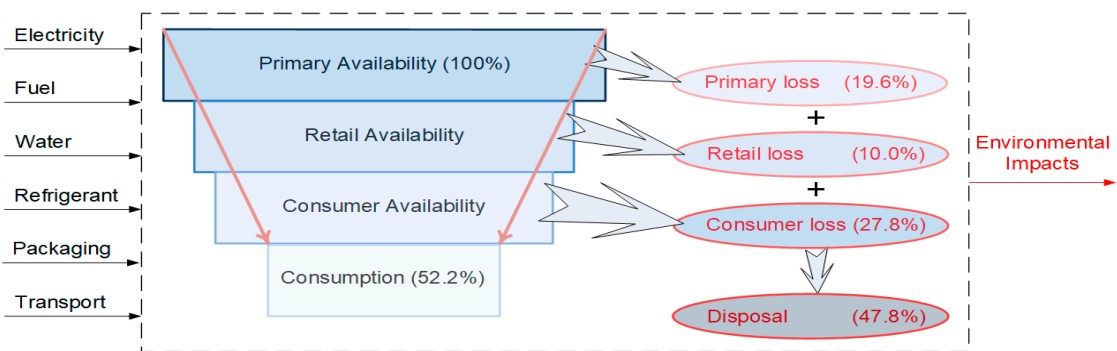

**Figure 1.** Schematic of food supply chain flows and associated losses.

*2.1. Goal and Scope*

The main goal of the study was to assess differences in the potential environmental impacts of dietary patterns of CFP and RFPs based on average annual consumption. The reference flow for this LCA is hence the cumulative amount of food consumed and associated losses. Results are expressed in kilograms per household per year stemming from activities along the entire supply chain and across all food commodity groups. The scope of the assessment is from the cradle-to-grave. The system boundary comprises the production, processing and packaging of food, transport through distribution networks, and storage at retail, consumption and disposal of the food waste and packaging materials. The environmental impact categories that were considered in the evaluation included categories for which impact characterization was available. Although the CEDA model [37] includes a wide range of potential environmental impact categories, we focused on those available using the TRACI 2.1 impact framework, as it has US-specific characterization factors [43]. The selected impact categories

were simulated using SimaPro© 8.4 [44]. The impact categories evaluated include the mid-point impact indicators such as global warming potential, eutrophication potential, acidification potential, ozone depletion, ecotoxicity, cancer and non-cancer health effects, smog formation potential, and fossil fuel consumption.

## 2.2. Tiered I/O-Based Hybrid Analysis

Due to the breadth of the retail sector in the I/O tables, which poses challenges to disaggregation into food commodity groups, we chose to link process models of the post-manufacturing supply chain to provide the cradle-to-grave perspective with commodity level granularity [45]. In the I/O part of the model, each food group was represented using a sectoral analysis based on the US Department of Commerce, Bureau of Economic Analysis [46] commodity groups. Subsectors were created by disaggregation so that each new I/O subsector had only one food group as its main product. Selective disaggregation of I/O sectors such as "fruit and vegetable canning, pickling, and drying; poultry and egg production; animal (except poultry) slaughtering, rendering, and processing" was conducted based on the subsector economic activity. Specifically, the disaggregation was based on a more detailed sectoral analysis available at the North American Industry Classification System (NAICS) level, because each of the I/O sectors represents a combination of several NAICS sectors [47]. Likewise, the ratio of NAICS subsector activity to the aggregated sector activity was used to disaggregate both inputs and outputs to the disaggregated subsectors. For example, when disaggregating the sector "fruit and vegetable canning, pickling and drying" into a fruit subsector and a vegetable subsector, the input value of "vegetable canning, pickling, and drying" to the new disaggregated subsector of "fruit canning, pickling, and drying" was set to zero in the revised I/O table. Additionally, the input of "fruit farming" was set to zero for the disaggregated subsector of "vegetable canning, pickling, and drying" process, and the input of "vegetable and melon farming" was set to zero for the disaggregated subsector of "fruit canning, pickling, and drying" process. These steps were necessary to avoid the double counting of raw materials used in the selected sectors. The reference flow of each disaggregated subsector was also assigned in proportion to its share of the original combined sector. The same concepts and procedures were applied to disaggregate other subsectors. Some food groups were assumed to be well approximated by the BEA commodity group with respect to production and manufacturing (Table S1 in Supplementary Material). The resulting input–output matrix was tested to ensure that the combination of the disaggregated categories provided the same result as the original, aggregated sector.

The I/O tables are enumerated in monetary form. Therefore, it was necessary to estimate the consumers' price and express the evaluation in the equivalent physical units to link with the post-production process-based models. The purchase price per kg of each food group was estimated using annual household expenditure characteristics according to the consumer expenditure survey [48]. The average number of people per household was 2.5, and the average annual expenditure for each food group was estimated by the combination of "food at home" expenditure and "food away from home" expenditure in proportion to the reported home and away expenditures (Table S2 in Supplementary Material). The calculated purchase price (retail price) was multiplied by the CEDA price conversion factor to obtain the producer's price, which was used in the computational platform to estimate the environmental burdens for the production and processing stage of each food group. These steps were needed because the I/O tables have emissions reported per dollar expended based on producers' prices. Since there are noticeable variations in the LCA studies of food products and because of the importance of global warming potential, GHGE results (from cradle to retail gate) per kg of each food group based on our approach are presented in Table S1 in Supplementary Material.

## 2.3. Uncertainty Analysis and Sensitivity Analysis

In any LCA study, it is important to understand the robustness of the conclusions. All the input data supporting the analysis have estimated mean values but carry a range of uncertainty. Sources of uncertainty can occur in various aspects including reliability of data, sample size relative to total

populations, representativeness of the sample, geographic variability, and many other characteristics. A quantitative analysis of the uncertainty due to the variability embedded in inventory data was carried out using SimaPro$^©$ 8.4 software. The ecoinvent pedigree matrix approach to assigning uncertainty to inputs was applied to unit processes generated from primary data. We assigned an intrinsic uncertainty estimate of 15% ($\sigma_G{}^2 = 1.15$) with the data quality pedigree scores of (1,3,2,1,1,2) for inventories of electricity, natural gas and refrigerants loss and (2,3,2,1,1,na) for all other input items including commodity price in the I/O model. All background unit processes taken from the ecoinvent database were adapted without changing the assigned uncertainty characterizations [49]. A Monte Carlo simulation (MCS) with 1000 iterations was used to propagate lifecycle inventory uncertainty to each impact category.

A sensitivity analysis was performed to determine the relative significance of commodity price on GHGEs, to provide information on how the outputs would change if the commodity price was changed. Since this study was based on the I/O modeling of economic value up to the retail gate, we carried out the analysis with a ± 10% change of commodity price, which is the most influential factor on concurrent results.

### 2.4. Limitations

Sustainability of food supply chain is guided by different factors, such as, investigating: (i) whether Short Food Supply Chain (SFSC) networks can outrank the alternative food supply chain—mainly to overcome the logistics-related impacts [50], (ii) influences due to changes in the consumers' behavior, as it was suggested through changes in the purchase behavior in real market scenarios [51], (iii) influences of increasing the efficiency of the food supply chain through technology improvements, particularly structural changes towards more efficient production [52]. However, these characteristics are not considered in the analysis, suggesting that additional research efforts to reduce perishable food loss through improved understanding of the role of preservation, reduced transit times, and possibly local, seasonal production are important to avoid the unintended environmental consequences of the recommended dietary shift.

## 3. Life Cycle Inventory

### 3.1. Food Supply Chain Inventory

We adopted the USDA ERS LAFA database as the basis for defining both food consumption and loss (including edible, inedible, avoidable and unavoidable losses) [40]. Although the data series is an accounting scheme as a proxy for food consumption based on disappearance data, it functions as an indirect estimation of food supply chain flows. The data series analyzes more than 200 commonly consumed food commodities, which are aggregated into food groups that match the USDA Food Patterns (FP) components used for dietary recommendations for a healthy diet based on the nutritional values of different foods. It accounts for most of the food consumed in the US. These commodities include vegetables, fruit/juices, milk/dairy, grains, red meat (i.e., beef, pork, veal, and lamb), poultry, eggs, fish/seafood, beans/peas, nuts/seeds, fats/oils, and sweeteners. One thing to note regarding this analysis is that consumption of alcoholic beverages and soft drinks is not evaluated because these are not recommended. The recommended USDA FP including vegetarian diets (e.g., RFP 2000 kcal, RFP 2600 kcal, RFP 2000 kcal vegetarian, and RFP 2600 kcal vegetarian) are calorie-based healthy dietary guidelines on how much Americans should eat of nutrient dense options from the major food groups and their subgroups, while placing limits on added sugars and solid fats [33]. The recommended average daily intake amounts at all calorie levels in USDA FP equivalents (e.g., cup of vegetables or a cup of fruit, ounces of meat, etc.) were converted to kilograms per household per year based on the reported serving equivalents, in order to estimate the household total food consumption on a weight basis. Table S3 in Supplementary Material presents the estimated annual food consumption associated with different dietary scenarios.

Food loss is broken down into losses at the primary level (post-farm production and processing), retail level (e.g., improper handling, food processing, food safety standards), and consumer level (e.g., losses from cooking and food preparation, excess food preparation, expired foods, spoilage, and plate waste). The food loss estimates for each food group are summarized in Section S1 and Table S4 in Supplementary Material. The database reports carcass weight for meat availability at primary level. We converted the carcass weight to live weight to account for non-edible losses at primary level. Since non-edible portions of animal products are used as by-products in various industries, the animal products were allocated into edible meat and by-products based on market values. When calculating the primary stage impacts, sector-specific economic census data were used to disaggregate the primary manufacturing stage based on revenue from primary products as a fraction of total sector revenue. Using USDA ERS data for primary and secondary product (by-product) revenue [53], for instance, we allocated 88.6% of the incoming burden of live beef to the boneless equivalent at the primary level and the remaining 11.4% of the incoming burden was allocated to by-products [54]. The allocation fraction between primary product and by-product of other food groups available in NAICS and USDA ERS data (USDA ERS data was used for red meat group because NAICS does not report disaggregated revenue data for each red meat) is presented in Table S5 in Supplementary Material.

Table 1 presents the total food consumption and losses, which aggregate to 1036 kg and 949 kg per household per year, respectively, over the whole life cycle of each food group for the CFP 2547 kcal diet. The cumulative losses represent 47.8% of the annual domestic food production, that is cumulative edible plus inedible loss, including byproducts, from farm-gate through consumption (Table S3 also presents other dietary scenarios). The aggregated totals increase to 1380 kg of projected food consumption and 1270 kg of food losses per year for the scenario of households adopting the USDA 2000 kcal dietary patterns (RFP 2000) assuming the same fractional loss rates for each food category. Recommended consumption for the vegetables, fruit/juices, milk/dairy, fish/seafood, and beans/peas categories is larger, while added sugars, fats and meats are lower compared to CFP. It is important to note that the projected consumption and losses under RFP 2000 kcal are higher than CFP, even though the caloric intake was reduced by approximately 20% and for an iso-caloric diet, the projected consumption and losses are even larger.

**Table 1.** Amount of production and consumption and associated loss of each food group per household per year based on current consumption patterns (CFP).

| Food Group | Production and Consumption of Current Diet (CFP 2547 kcal) (kg/household/year) | | Food Loss in Each Supply Chain Stage Based on Current Diet (CFP 2547 kcal) (kg/household/year) | | | |
|---|---|---|---|---|---|---|
| | Production | Consumption | Primary Loss | Retail Loss | Consumer Loss | Total Loss |
| Vegetables | 442 | 180 | 141 | 25.6 | 96.0[b] | 263 |
| Fruit and Juices | 290 | 134 | 57.4 | 21.5 | 77.2[b] | 156 |
| Milk and Dairy | 292 | 201 | 0.322 | 32.6 | 58.3 | 91.1 |
| Grains | 222 | 153 | 0 | 26.6 | 42.5[b] | 69.1 |
| Red meat | 261[a] | 87.5 | 141 | 5.38 | 27.5 | 174 |
| Poultry | 129[a] | 63.5 | 47.7 | 3.15 | 14.7 | 65.5 |
| Eggs | 36.8 | 21.4 | 0.552 | 3.26 | 11.5 | 15.3 |
| Fish and Seafood | 18.1 | 11.0 | 0 | 1.48 | 5.65 | 7.13 |
| Beans and Peas | 7.79 | 6.59 | 0 | 0.467 | 0.732 | 1.20 |
| Nuts and Seeds | 12.4 | 10.6 | 0 | 0.742 | 1.07 | 1.81 |
| Fats and Oils | 113 | 72.2 | 0.157 | 21.7 | 18.8 | 40.6 |
| Sweeteners | 161 | 96.5 | 0 | 17.7 | 46.7 | 64.4 |
| Total | 1985 | 1036 | 388 | 160 | 401 | 949 |

[a] The production of red meat and poultry represents live weight. Meat-specific conversion factors were used to convert carcass weight to live weight (beef: 63%, pork: 72%, other red meat: 63%, poultry: 58%). [b] Nonedible share is included as loss.

*3.2. Retail*

Retail is a highly concentrated industry, which has substantial input flows [55]. Retail stores consume significant energy and resources that contribute to supply chain environmental impacts. The major contributing activities are the electricity for store operations (overhead) and refrigeration system, loss of refrigerants due to leakage, natural gas consumption, and water usage [56,57]. Data on the sales volume and information of space occupancy (refrigerated versus non-refrigerated) were analyzed to determine the burdens assigned for each food group. The allocation of burdens for each food group was calculated using household expenditure data [48] (Table S6 in Supplementary Material). The retail impacts for each refrigerated food group were assigned based on a combination of the refrigeration and overhead (including air-conditioning and lighting, etc.) burdens. Each non-refrigerated food group was assigned impacts (based only on store-wide overhead burdens) through allocation based on consumer expenditures.

Table 2 presents the reference data of a typical grocery retail outlet for the vegetables group as an example. Non-refrigerated food groups have similar reference data except for the refrigeration burden. The consumption of electricity, natural gas, and water as well as the annual refrigerant loss was coupled with data published by ASHRAE [58] to generate estimates of the burden of building operations. Direct expansion refrigerant systems are commonly used [59] and typically loaded with R-22, R-404A, and R-507A; for mixtures, the composition of the mixture was used to determine the appropriate global warming potential [60]. These systems have a compressor and the refrigerant is pumped through a pipe network that is the source of most refrigerant loss, generally due to catastrophic events (e.g., a broken pipe). There is a typical load of 1590 kg of refrigerant and an average annual leak rate of 20% [61]. Stand-alone refrigeration equipment has a relatively small refrigerant charge and leak rate, thus it was not accounted [62].

**Table 2.** Reference data of refrigerated food group at a typical supermarket retail outlet. This table is for the vegetables group.

| Composition | Symbol | Amount | Unit | Note |
|---|---|---|---|---|
| Total grocery store area | $A_{G,T}$ | 4270 | $m^2$ | FMI, Supermarket facts |
| Electricity usage | $C_{E,A}$ | 557 | kWh/$m^2$/year | ASHRAE 2012 Handbook |
| Overhead demand fraction | $D_{E,O}$ | 56 | % | Energy Star, Building upgrade manual |
| Overhead allocation fraction | $FS_{V,T}$ | 5.3 | % | Consumer expenditure survey |
| Refrigeration demand fraction | $D_{E,R}$ | 44 | % | Energy Star, Building upgrade manual |
| Refrigerated allocation fraction | $FS_{V,R}$ | 10.2 | % | Consumer expenditure survey |
| Natural gas usage | $C_{N,A}$ | 15.3 | $m^3$/$m^2$/year | ASHRAE 2012 Handbook |
| Overhead demand | $D_{N,O}$ | 87 | % | Energy Star, Building upgrade manual |
| Refrigerant load | $L_{R,T}$ | 1590 | kg | US EPA, Supermarket report |
| Annual leak rate | $LR_{R,A}$ | 20 | % | US EPA, Supermarket report |
| Water consumption | $C_{W,A}$ | 2880 | liter/$m^2$/year | Aquacraft Inc. report |

Based on this information, electricity, $V_{E,G}$, and refrigerant loss, $V_{R,G}$, burden of the vegetables group at a typical supermarket was estimated (terms defined in Tables 2 and 3) from the following equations, respectively:

$$V_{E,G} = C_{E,A} \times A_{G,T} \times (D_{E,O} \times FS_{V,T} + D_{E,R} \times FS_{V,R}) \tag{1}$$

$$V_{R,G} = L_{R,T} \times LR_{R,A} \times FS_{V,R} \tag{2}$$

**Table 3.** Calculated burdens of vegetables group in a typical supermarket and allocated burdens of vegetables group per kg displayed.

| Resource | Symbol | Annual Burden of Vegetables in a Typical Supermarket | | Symbol | Burden per kg of Vegetables Displayed | |
|---|---|---|---|---|---|---|
| Electricity | $V_{E,G}$ | 173,808 | kWh/year | $V_{E,M}$ | $2.58 \times 10^{-1}$ | kWh/kg |
| Natural gas | $V_{N,G}$ | 3018 | m$^3$/year | $V_{N,M}$ | $4.48 \times 10^{-3}$ | m$^3$/kg |
| Refrigerant | $V_{R,G}$ | 32.5 | kg/year | $V_{R,M}$ | $4.82 \times 10^{-5}$ | kg/kg |
| Water | $V_{W,G}$ | $5.67 \times 10^5$ | liter/year | $V_{W,M}$ | 0.842 | liter/kg |

Then, the electricity consumption per kg of vegetables was calculated based on a national average supermarket store area (Table S7) [63], along with the average quantity of vegetables displayed per year (estimated from LAFA data of consumer purchases plus retail losses). Similar procedures were applied to calculate refrigerant loss, natural gas, and water usage burdens (Section S2). The estimated burdens for other food groups are presented in Table S8 in Supplementary Material.

*3.3. Consumption*

The resources used at the consumer phase including transportation for shopping trips, refrigeration, food preparation, dishwashing, and waste treatment were analyzed. We allocated the resources usage burden including electricity, which is the highest impact driver at consumer stage, to each food group based on consumer food expenditure data (Table S6). According to the Food Marketing Institute report, the average US household made 104 trips for grocery shopping annually [64]. According to the National Household Travel Survey (NHTS), the average vehicle roundtrip distance for all-purpose shopping was 10.3 kilometers [65]. In this study, we assumed that the grocery shopping distance is equivalent to the all-purpose shopping distance. The average annual household grocery purchase and expenditure for each food group was used to allocate passenger car distance traveled for grocery shopping to each commodity group. The passenger car transportation distance allocated for the vegetables group, as an example, was estimated to be 0.179 km per kg purchased. An equation and all other food group estimations for passenger car transportation distance are presented in Section S3 and Table S9 in Supplementary Material.

The US EIA Residential Energy Consumption Survey (RECS) reports average annual energy use for home refrigeration to be approximately 1250 kWh [66]. The home refrigeration attributable to each refrigerated food group was calculated based on the same allocation method as supermarket refrigeration. On a commodity group basis, this approach will slightly overestimate in-home refrigeration because the fractional space occupied should be reduced by items like ketchup and salad dressing which are not typically refrigerated in grocery stores. Refrigeration energy usage per kg of products stored at the household refrigerator are shown in Table S9. For vegetables, it was estimated to be 0.552 kWh/kg.

According to the Umatilla Electric Corp. report [67], food preparation appliances, including a range with an oven, microwave, dishwasher, etc., consume 1920 kWh per household per year (Table S10). For most of the cooking appliances, we adopted the allocation scheme based on the fraction of expenditure per household rather than disaggregating the energy consumption of cooking appliances to each food group because of the complexity of cooking. Resource usage by specific cooking appliances, which are used for only a certain food category, were assigned solely to the specific food group. For instance, electricity usage of a toaster was assigned solely to the grains food group. Natural gas and water usage for food preparation are not included in this analysis; data are not available on the fraction of cooking appliances, therefore we assumed that electric ovens and stoves are representative for energy consumption and water usage for cooking is minimal. Electricity consumption for food preparation per kg of vegetables was estimated to be 1.31 kWh/kg (Table S9). Details of energy consumption for preparing food in different appliances are shown in Table S10.

Detergent and water burdens for dishwashing were estimated, with information taken from the Energy Star criteria for a standard sized dishwasher model [68]. A standard sized dishwasher, holding eight place settings and six serving pieces, is measured to use 22.0 L of water per cycle. In this study, it was assumed that an average household would operate a dishwasher once per day. For each dishwashing cycle, 25 g of detergent (from the product label) is used. Instead of attempting to estimate the dishwashing burden based on the allocation of cookware and tableware to each food group, it was allocated based on the fraction of expenditure per household in each food group.

## 3.4. Post-Consumer Waste Treatment

In the US, total municipal solid waste (MSW) generated in 2010 was approximately 251 million tons [69]. It was reported that about 33% of the MSW is packaging waste [70] and food packaging waste accounts for two thirds of the packaging waste [71]. This ratio was used in estimating the amount of food packaging disposed. Marsh and Bugusu (2007) provided information on food packaging materials and uses in Table 3 of their study [70]. Additionally, Table S11 presents the types of packaging materials used for each food group obtained from various studies [72–74]. When possible, we assigned specific packaging materials to specific food groups. For more generic packaging materials such as corrugated boxes, plastic bags, wood pallets, and aluminum (foil), use was assigned to each food group in proportion to the packaged quantity from the processing stage. Table S12 presents the type and estimated amount of packaging materials used for each food group. The amount of packaging material recovered was incorporated into modelling as recycling rate for each type of food packaging material. For example, the recycling rates of packaging materials are paper and paperboards (59%), glass (25%), and plastics (9.4%). The remaining portion was assumed to be disposed of at landfill sites [70]. We modeled waste scenarios using the ecoinvent dataset for the disposal of wasted food and packaging materials [75]. According to recent data from the US Environmental Protection Agency [76], more than 96% of food waste is landfilled, while 2.5% is composted. There is no certain data for food waste incineration. Due to the lack of information available and the dominant role of landfilling, we chose a landfilling unit process to model the disposal phase of solid food waste. Milk/dairy and fruit/juices groups contain liquid and solid waste streams. We allocated liquid and solid waste streams in the milk/dairy group to municipal wastewater treatment and landfill, respectively, based on the liquid (fluid products) fraction versus solid (cheese, butter, etc.) fraction of 78.6% and 21.4%, respectively. In the fruit/juices group, the ratio is 25.8% liquid waste and 74.2% solid waste [39].

## 4. Results

### 4.1. Life Cycle Impact Assessment

Figure 2 presents GHGEs as a function of total food consumption and loss for each food group associated with CFP and RFP recommendations. Five columns in each food group indicate dietary patterns in the order of current food pattern (CFP 2547 kcal) and USDA recommended food patterns (RFP 2000 kcal, RFP 2600 kcal, RFP vegetarian 2000 kcal, and RFP vegetarian 2600 kcal). The dotted lines represent cumulative GHGEs on the secondary Y-axis. CFP contributes 8.80 tons of carbon dioxide equivalent ($CO_2$-eq) emissions per household per year. This corresponds to 9.64 kg $CO_2$-eq per person per day. It increases to 9.61 tons $CO_2$-eq emissions per household per year for the RFP 2000 kcal recommendation, which corresponds to 10.5 kg $CO_2$-eq per person per day. Vegetables, fruit/juices, milk/dairy, and fish/seafood were the major contributors of the increased GHGEs, due to greater recommended intake and associated losses. The levels of food loss/waste of different food groups had a substantial role in deriving the results. This study also contains granular post manufacturing information, encompassing a detailed estimation of retail and consumer phases. Furthermore, EIO-LCA results are sensitive to product price. These factors resulted in a slightly higher GHGE in our study compared to other EIO-LCA-based food system studies [17,77]. One thing to note from the figure is that the recommended vegetarian diets (RFP 2000 kcal vegetarian and RFP 2600 kcal vegetarian) containing

no red meat, poultry or fish/seafood tend to have the lowest GHGE impact. For the dietary scenarios, the GHGEs of RFP 2000 kcal vegetarian and RFP 2600 kcal vegetarian was 22% and 11% lower than CFP 2547 kcal diet, respectively. Under CFP, the red meat group, including loss, was the single largest GHGEs contributor (41.5% or 3.65-tons $CO_2$-eq emissions per household per year). Based on the RFP 2000 scenario, emissions associated with the milk/dairy (24.1%) and red meat (20.0%) groups were the two major GHGE contributors, followed by the fruit/juices group (19.3%). To maintain a constant calorie content of 2000 kcal, in the case of reductions in animal based products, the RFP increases the share of other foods: 62% higher intake of vegetables, 155% increase of fruit/juices, 98% increase of dairy products, and 171% increase of fish/seafood, compared to the respective consumption in the CFP. Such changes in the dietary patterns as well as GHGE intensities of the food groups (kg $CO_2$-eq emissions per kg product) affected the results. For example, the red meat group emits 3.65 kg $CO_2$-eq for the CFP 2547, while the reduced intake of red meat in RFP 2000 emits 1.9 kg $CO_2$-eq per household per year, corresponding to a 47% reduction in the GHGEs. Meanwhile, the increased intake of fish/seafood in RFP 2000 increases 171% of GHGEs compared to the representative CFP. Likewise, the amount of intake coupled with GHGEs intensities of other food groups had an effect on the results. The cumulative GHGEs of vegetables, fruit/juices, and milk/dairy groups in the CFP 2000 were 2.87 kg $CO_2$-eq per household per year, while the emissions from these groups were responsible for 5.74 kg $CO_2$-eq per household per year under the RFP 2000 (details are presented in Supplementary Material, Table S13). In addition, when the total consumption of the RFP food commodities, along with the respective losses, was compared to the CFP case, it is evident that more food losses are associated with RFP, which is driven by higher post-harvest losses (mainly at retail and consumer phases) for the plant-based products. Under the CFP 2547 kcal scenario, relative contribution to the total GHGEs from the primary production (comprising both agriculture production and processing) was 72%, followed by the retail (8%) and consumption (20%) stages. The contribution from the respective stages of the supply chain, along with the changes in the food consumption in the RFP 2000 kcal, reduced to 67% of the related GHGEs at primary level. However, increments were observed at both the retail (10%) and the consumption (23%) stages.

The LAFA database reports food availability for each meat class, from which both the specific availability and the loss information for each meat type were accounted. In the red meat group, beef is the dominant GHGEs driver because of its larger emission intensity; it contributes 75.1% of the red meat group, which is equivalent to 31.1% of total emissions, while it accounts for only 6.38% and 4.95% of food consumption by calorie and weight, respectively. The contribution assigned to the beef supply chain drops to 15.3% of total GHGEs under the RFP 2000; which accounts for 2.0% of food consumption by weight. Thus, despite the anticipated benefits of the recommended diet, GHGEs associated with the food system are not reduced by a shift to align with RFP 2000 dietary guidelines, and there is an increase of 9.17% compared to CFP. A similar conclusion, with a lower fractional increase than this study, is shown by Birney et al. (2017) and Tom et al. (2016) with three different dietary scenarios [11,19]. In an iso-calorific shift from the current US diets to USDA dietary recommendation, Heller and Keoleian (2015) reported a 12% increase in diet-related GHGEs [14]. Similar to the results obtained in the current study, they also suggested that even with more than 20% reduction in caloric intake and a considerable reduction in meat consumption, a shift to a recommended diet may not lead to a significant decrease in the GHGEs, even though the study did not account for food losses [14]. Further comparison with other studies is discussed in Section 5.

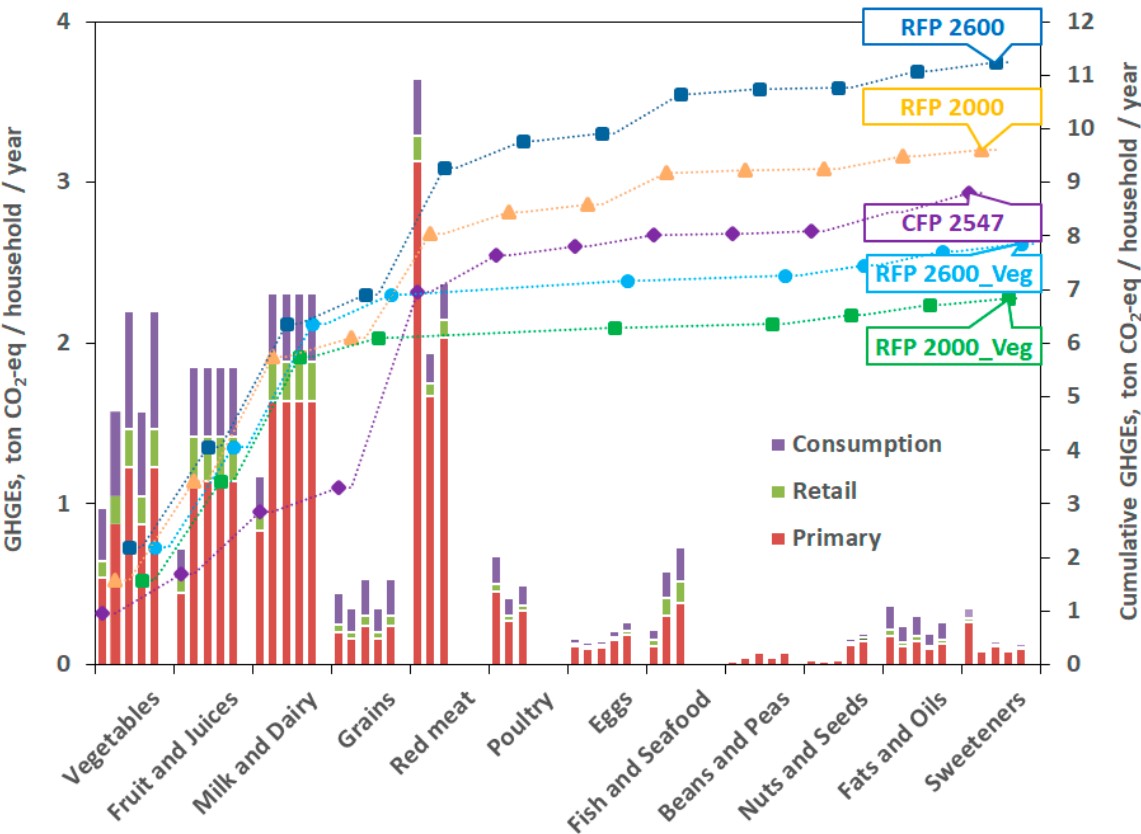

**Figure 2.** GHGEs associated with current and recommended dietary patterns, including food loss/waste.

Figure 3 presents the relative contribution of each food group within the five dietary patterns to various environmental impact categories, using CFP as a 100% reference (Table S14 presents numerical results). The legend shown in the right side for each food group should be read from bottom to top, corresponding to the pattern in the stacked column chart. Because the units of measurement for the individual impact category are different, results are presented as a contribution analysis for each impact category. Under the CFP, red meat consumption and loss was the largest impact driver for most impact categories apart from eutrophication potential, to which the poultry group contributes the most. The fruit/juices group was the largest contributor to ozone depletion impact under the RFP 2000 arising from refrigerant loss. As the chart displays, the RFP 2000 scenario, compared to the CFP, has higher footprints for most of the impact categories except for smog, acidification and eutrophication, as a result of the shifts in quantities consumed. The recommended reduction of red meat consumption decreases the overall impacts of those three categories for RFP. Heavy metals emissions associated with landfill disposal of food waste and packaging materials drive the increased carcinogenic impact due to the sharing of burdens from all municipal waste disposed in landfills. The emissions from the electricity supply chain and combustion of fossil fuels are the next largest contributors. For ecotoxicity, heavy metals emissions associated with fertilizers and waste disposal processes are the major contributors. Under the recommended vegetarian diets, most of the impact categories were lower than the reference, but ozone depletion, human toxicity, and ecotoxicity impacts increased. The increased intake of vegetables, fruit/juices, and milk/dairy groups explains these results. Based on the results of this study, selection of a sustainable diet is not simple as there are trade-offs among environmental impacts.

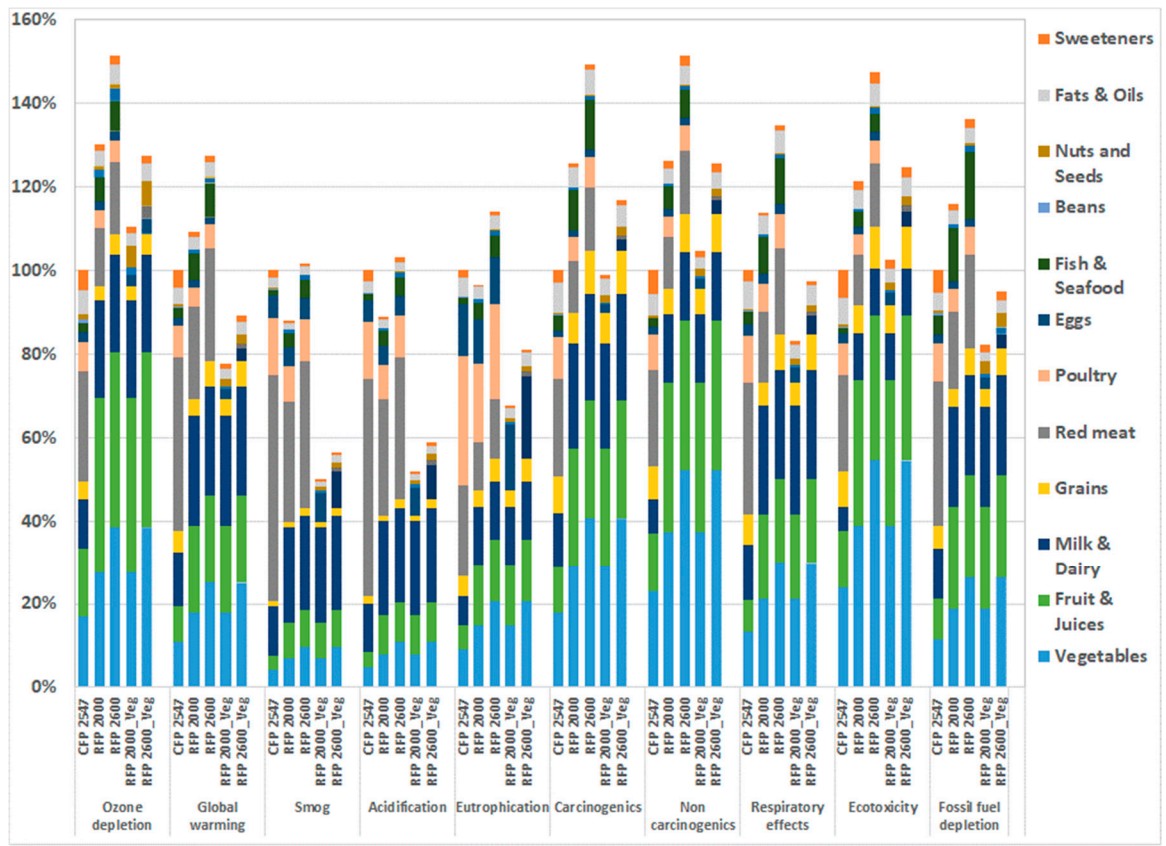

**Figure 3.** Relative contribution of each food group to environmental impacts associated with five food consumption patterns.

## 4.2. Normalization

Normalization is a useful step for interpreting the relative significance of the various environmental impact categories under study. It can provide guidance to support target activities on those impacts where significant benefits can be achieved. The overall emissions contributing to each impact category were estimated and then normalized to a per household basis for US-specific conditions [43]. As shown in Figure 4 for CFP analysis, the normalization results indicate that ecotoxicity is the highest normalized impact category, thus improvement in this category will lead to larger relative reductions. The major contributing substances were heavy metals leached to river/groundwater and soil resulting from municipal waste management, crop field operations, and the disposal of coal mining tailings associated with the production of electricity. The ecotoxicity and human toxicity impacts appear to be unreasonably high (over 100% of an average household's annual contribution) because of the emission inventory of heavy metals in background data. We tested an alternative waste management unit process by replacing the ecoinvent municipal waste unit process with the European Life Cycle Database (ELCD, v3.2) unit process [78] and the result shows that the impacts in ecotoxicity and human toxicity are about twice as low. This is a manifestation of the well-documented uncertainty in heavy metals emissions and impact characterization factors. Human toxicity and eutrophication are the next highest impact categories. From a consumer perspective, these impacts can be mitigated substantially by reducing food waste and electricity consumption. Based on our analysis as presented in Figure 4, food production and consumption are responsible for approximately 14.5% of the annual US global warming potential, mostly driven by red meat (6.0%), milk/dairy (1.9%), and vegetables (1.6%).

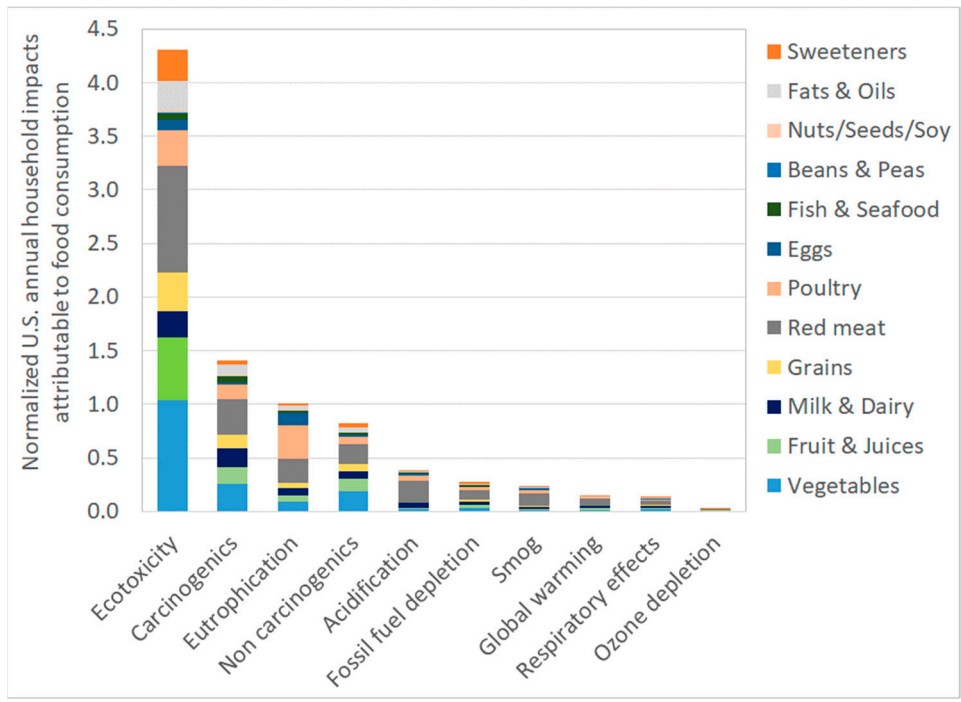

**Figure 4.** Normalized cradle-to-grave impacts for current US food consumption patterns.

*4.3. Uncertainty Analysis*

Uncertainty analysis was performed to analyze GHGEs associated with five alternative dietary patterns, to characterize the robustness of our conclusions regarding differences in potential environmental impacts. Figure 5 presents the results of 1000 MCS runs. The numerical results of GHGEs and other impact categories are presented in Table S15 in Supplementary Material. While the figure suggests that the differences in emissions across the dietary patterns are relatively small due to the overlapping error bars, it can be demonstrated that some of these dietary patterns are statistically better or worse than others on GHGE impact. It confirms that an iso-caloric estimation of the RFP 2600 scenario (mean value of 11.2 with a range from 10.3 to 12.4 tons $CO_2$-eq per household per year) shows a probability of having higher GHGEs than CFP 2547 (mean value of 8.81 with a range from 7.89 to 9.95 tons $CO_2$-eq per household per year) with 95% confidence. As well, RFP 2000 (mean value of 9.61 with a range from 8.79 to 10.6 tons $CO_2$-eq per household per year) demonstrates the likelihood of having higher GHGEs than CFP 2547 overall, but there is no difference in GHGEs between these two dietary patterns at the primary level. Cumulative emissions associated with different levels of food consumption coupled with different levels of food loss/waste at retail and consumer phases played a significant role leading to this result. It also confirms that the recommended vegetarian diets (RFP 2000 kcal vegetarian and RFP 2600 kcal vegetarian) containing no red meat, poultry or fish/seafood have the lowest GHG emissions (mean values of 6.82 and 7.84 with 95% confidence intervals of 6.17–7.53 and 7.13–8.66 tons $CO_2$-eq per household per year, respectively). Additional interpretations about the findings are discussed in Section 5. This analysis, based on the accounting of emissions over the full life cycle of food consumption patterns coupled with different levels of food loss/waste at retail and consumer phases, provides an insight into the relative importance of post-agricultural stages and information to support the policymaking of dietary guidelines.

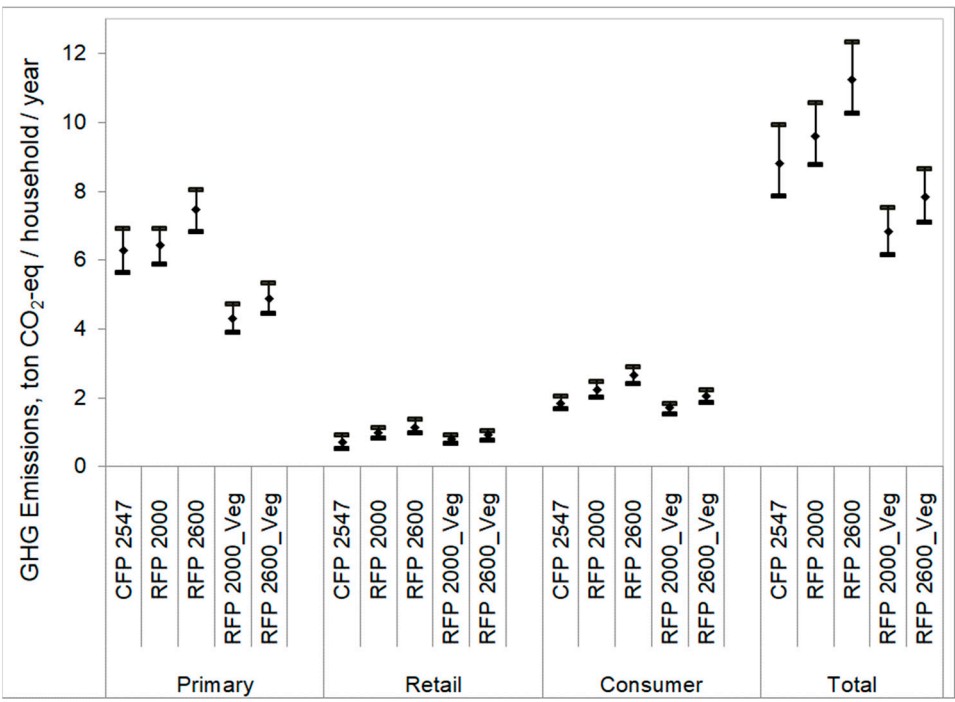

**Figure 5.** Results of 1000 Monte Carlo runs for uncertainty analysis of dietary patterns. The error bars represent the 95% confidence interval of the mean.

### 4.4. Sensitivity Analysis

Figure 6 presents fractional change in GHGEs associated with a 10% overestimate/underestimate of commodity price. Under CFP, a 10% increase in beef price results in approximately a 3.0% increase in GHGEs (Figure 6a), due to the relatively large contribution of beef to the overall footprint and that, because the burden is linked to price, a higher price implies higher GHGs entering the retail and consumption phases. A 10% increase in milk/dairy price increases GHGEs by 1%, which is the second most sensitive driver to GHGEs. Under RFP 2000 dietary patterns, milk/dairy price becomes the most sensitive driver (1.7%) to GHGEs followed by beef (1.4%) and fruit/juices (1.1%) (Figure 6b). Consequently, the commodity price clearly affects the numerical results, but it does not affect the overall interpretation of the result.

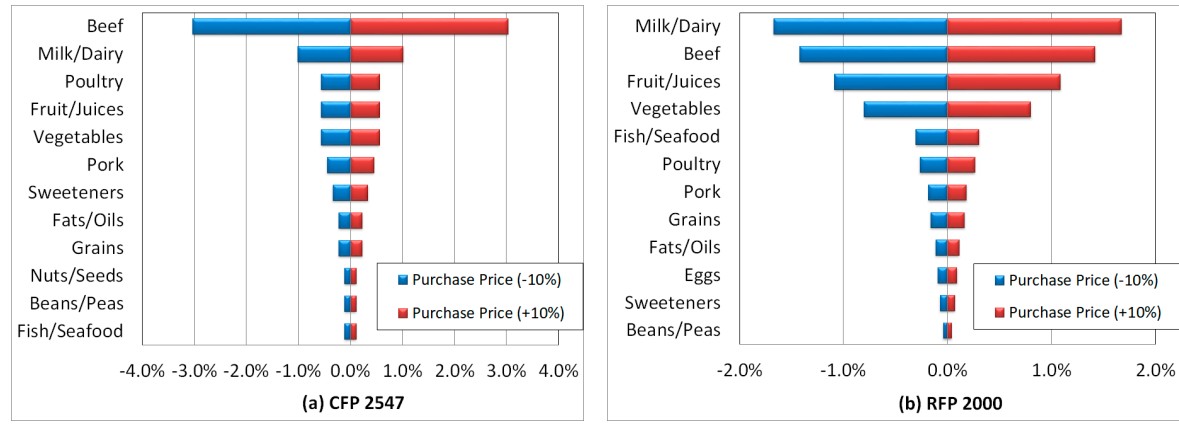

**Figure 6.** Fractional change in GHGEs associated with a 10% overestimate/underestimate of commodity price using CFP 2547 and RFP 2000 dietary patterns.

The LAFA dataset accounts for inedible portions of animal products (bones) as losses at the primary level and presents values at consecutive stages as boneless equivalents. Inedible portions of processed fruits and vegetables (peels, cores, etc.) are also accounted as losses at the primary level. These losses are classified as by-products at the primary level as described in Section 3.1. However, inedible portions of fresh fruits and vegetables are accounted as losses at the consumer level. Given the economic basis of this assessment, we assumed that the inedible portions of fruits/vegetables carry the same economic value as edible portions as consumers are buying whole fruits, peel/core and all. We carried out a sensitivity analysis by applying that 16% of fruits and 12% of vegetables are inedible at the consumer level [11], and carry no economic value. It decreases GHGE by 1.0% and 2.0% under CFP and RFP 2000, respectively.

## 5. Discussion

### 5.1. Environmental Impacts and Mitigation Potential

It is important to note that the projected consumption and losses under RFP 2000 are higher than CFP 2547, even though the caloric intake was reduced by approximately 20%. For an iso-caloric diet, the projected consumption and losses are even larger. The recommended reductions in the consumption of red meat, poultry, grains, eggs, fats/oils, and sweeteners and associated losses decrease GHGEs, but this decrease was only possible due to a significant increase in the consumption of vegetables, fruit/juices, milk/dairy, fish/seafood, and beans/peas products, hence increasing the share of their impacts in the total food supply chain under the alternate consumption scenarios. However, for the vegetarian diets (without animal sourced foods) for both the current and the recommended scenarios, the potential environmental impacts were lower, as discussed in Section 4.1. Furthermore, the majority of previous studies concluded that changing dietary patterns to recommended diets can decrease the potential environmental impacts [24,79–81]. A partial explanation for different conclusions in this study is related to differences in the current US consumption patterns compared to the base case in other studies. For instance, in Meier and Christen (2013)'s study for German dietary patterns [80], the recommended dietary guidelines accounted a marginal increase in the intakes of vegetables and milk/dairy products and larger reductions of intakes in red meat, poultry, and egg groups compared to those recommended in the USDA Dietary Guidelines. It also recommended a reduced consumption of fruits, which is the opposite of the USDA's recommendation. Green et al. (2015) adopted an average optimized diet among UK adults [79] adapted to World Health Organization (WHO) dietary recommendations, which accounted a reduced milk/dairy products consumption and smaller increments in fruit/juices and fish/seafood consumption, which was different to what was recommended in the USDA Dietary Guidelines. In addition, regional dissimilarities in the agricultural production system may also partially explain the differences. In our study, retail and consumption stages coupled with food loss/waste contributed significantly to the overall potential environmental impact. For example, the summation of the retail and consumption stages contributed 28% and 33% of the overall GHGEs under CFP and RFP 2000, respectively.

Across all the food groups, food losses contributed a significant amount to GHGEs. Food losses at the retail and consumer levels represent most of the avoidable food waste, except for non-avoidable shrinkage during cooking. Potential mitigation strategies to reduce the environmental burdens can be achieved, mainly through the integration of (i) food supply chain optimization possibilities, e.g., through increased productivity and reduced waste, (ii) improvement in the supply chain efficiency through technology enhancement and behavioral changes [82], and (iii) judicious management of generated waste to minimize the undesired emission from landfills. However, this is likely possible only with the implementation of stringent policy measures. García-Muros et al. (2017) argued that "carbon-based food taxes" imposed on food choices would not only help to reduce emissions but also motivate consumers to change consumption patterns towards healthier diets [83]. Other potential strategies for minimizing the food losses can be achieved by improved labeling and food storage,

reducing the transit times between the food supply chain and nudging consumer behavior [84]. From the standpoint of waste management, composting and anaerobic conversion can mitigate GHG emissions. Furthermore, alternative recycling of food waste as livestock feed was also reported for lowering environmental impacts [85]. The most important conclusion from the study is that without taking care in the implementation of dietary shifts, the unintended consequences on the environmental effects of food supply may be oppositely directed to our goals. The implications of these results to decisions made by individuals or policymakers intending to influence the environmental sustainability through dietary modification further underscore the importance of careful management of food loss in supply chains.

### 5.2. Strengths and Limitations of the Current Study

Since the scope of the study was to evaluate the impacts of different food patterns of the US, it was necessary to select a reference flow equivalent to the total food supply chain, in terms of kg of food consumed at household level, including losses. For the CFP food pattern, the required caloric intake was 2547 kcal, whilst it was 2000 kcal in the RFP. On an absolute comparative analysis, this might be regarded as comparison of "apple and orange" but, as stated above, the evaluation helped to investigate the potential environmental impacts of the food supply chain that consumers will eventually receive for consumption at household level. Kendall and Brodt (2014) also compared various food products, where they used different functional units, e.g., mass, serving size, energy content, protein content, and a composite nutrient score [86]. Studies published on food mix showed varying environmental impacts depending on the choice of the functional unit [87]. Nevertheless, the function of food, in addition to nutrition, is also characterized for qualities/services that they provide, e.g., varying taste, aesthetics, impact on health and economy, etc., which are indeed difficult to capture in a single functional unit [18].

In this study, a process-based LCA model is applied for the retail and consumer phases. In CEDA, the direct requirement matrix does not include product-specific impacts of the post-manufacturing supply chain. Even though many sectors do purchase from retail services and wholesale services, the precise nature of these purchases is unknown. A sector may purchase ordinary goods or services through retail or wholesale as an input to production, or a sector may purchase the sales and logistics services from retail or wholesale services to sell their product. The current I/O table tries to eliminate the latter contribution, but the table may still contain such activities. There is no quantitative assessment on the effect of the accounting of these activities in an I/O table, and it remains as a potential source of error [88]. The hybrid I/O modeling may thus add a few or so percent of double counting at retail phase assessment, but it will not affect the overall results. In addition, because we disaggregated several of the food sectors based on NAICS data, which are not readily available for the value-added matrix, the retail stage burden could not be allocated with the desired granularity, thus we adopted process-based LCA at retail and consumer phases for better resolution.

The analysis of household level should not be extrapolated to national level because underlying assumptions of the I/O model and LCA would be violated. Large-scale changes in consumption patterns will affect the amount of production of different commodities and could disrupt the whole food supply chain system. National scale modeling should be based on a more complex evaluation of elasticities through general equilibrium modeling of the economy, to project the potentially large shifts in sector activity which might result from large-scale adoption of alternative dietary patterns and thus changes in environmental impacts.

It is widely argued that among the numerous factors contributing to the current US obesity problems, consumption of higher calorie and unhealthy diets are the most common causes. If the environmental impacts of dietary patterns are to be judiciously evaluated, then it is important that impacts are analyzed with respect to the total calorie and nutritional values that each food category provides. In such cases, it is recommended that consequences to health impact due to consumption of the both lower calorie foods with considerable low fats and meat products are analyzed [13]. A limitation

of this work is that the health benefits of RFP are not included in the assessment. In particular, the contribution of certain micronutrients from animal source foods is inadequately accounted.

## 6. Conclusions

A life cycle assessment of different dietary scenarios was conducted using a tiered hybrid input–output method. A cradle-to-grave evaluation of the food supply chain was carried out for the entire food system under the current US consumption patterns and the recommended USDA food consumption patterns. Unlike adopting a vegetarian diet, adoption of the USDA dietary guidelines has a probability of increasing GHGEs and other environmental impacts. In this study, the retail and consumption stages coupled with food loss/waste contribute significantly to the higher emissions results of recommended consumption patterns. The recommended reductions in consumption of red meat, poultry, grains, eggs, fats/oils, and sweeteners and associated losses decrease GHGEs, but this decrease is offset by substantial increases in vegetables, fruit/juices, milk/dairy, fish/seafood, and beans/peas products consumption and emissions. Food losses at the retail and consumer levels represent most of the avoidable food waste, except for non-avoidable shrinkage during cooking. Avoidable food loss in the consumer phase can be reduced by changes in consumers' behavior. The overall outcome highlights the importance of incorporating environmental considerations, with a full accounting of the effects of food loss/waste as sustainability deliberations, into dietary guidelines. The current study also highlights the need for additional studies, particularly oriented toward accounting for potential unintended consequences resulting from large-scale shifts in food patterns.

**Supplementary Materials:** The following file is available online at http://www.mdpi.com/2071-1050/12/4/1586/s1, Table S1: EIO-LCA commodity sector mapping for each food group, Table S2: Expenditure in US dollar per household per year and average purchase price per kg of each food group. CEDA price conversion factor is multiplied by the purchase price to obtain producer's price, Table S3: Estimated annual food consumption per household associated with dietary patterns among the US population, Section S1: Calculation of total food loss using USDA ERS loss-adjusted food availability (LAFA) database, Table S4: Percent of food loss at each supply chain using USDA ERS loss-adjusted food availability database, Table S5: The allocation fraction between primary and secondary product (by-product) by NAICS revenue data and USDA ERS data, Table S6: Food group allocation fraction for the estimation of space share of each food group at a typical supermarket and environmental burdens at retail phase and consumer phase, Table S7: Store counts by grocery channels, Section S2: Calculation of retail burdens for the vegetables group at a typical supermarket outlet, Table S8: Allocated burdens in a typical supermarket and allocated burdens per kg displayed across all other food groups, Section S3: Calculation of consumer phase for the vegetables group as an example, Table S9: Energy consumption per kilogram of commodities in the consumer phase across all food groups, Table S10: Electricity consumption of food preparation appliances for each food group per household per year. The allocation was based on the household expenditure data, Table S11: Type of packaging materials used for each food group, Table S12: Food packaging types and estimated amount of packaging materials used for each good group in million tons, Table S13: Relative contribution of the food groups to the environmental impact, e.g. GHGEs under the CFP 2547 kcal and the RFP 2000 kcal scenarios, Table S14: Relative contribution of each food group to environmental impacts associated with dietary patterns, and Table S15: Results of 1000 Monte Carlo runs for uncertainty analysis associated with dietary patterns.

**Author Contributions:** G.J.T. developed analytical concepts and contributed to critical revision; R.P. supported the methodological approaches and manuscript preparation. D.K. performed the LCA. and drafted the manuscript. All authors have read and agreed to the published version of the manuscript.

**Funding:** This research was partially funded by the National Cattlemen's Beef Association (Grant #: 1406-R, 2014).

**Acknowledgments:** The authors acknowledge the association for the financial support and hypothesis development. We also thank senior scientists, Carolyn Scrafford, Leila Barraj and Erin Barrett (Exponent, Inc., 1150 Connecticut Ave., NW Suite 1110, Washington, DC 20036, USA), for supporting data analysis as well as Drs. Shalene McNeill, Kim Stackhouse-Lawson and Sara Place (National Cattlemen's Beef Association, 9110 E. Nichols Ave., Suite 300, Centennial, CO 80112, USA) for invaluable comments.

**Conflicts of Interest:** The authors declare no conflict of interest. The funders had no role in the design of the study; in the collection, analyses, or interpretation of data; in the writing of the manuscript, or in the decision to publish the results.

**Abbreviations**

| | |
|---|---|
| ASHRAE | American Society of Heating, Refrigerating, and Air-Conditioning Engineers |
| BEA | Bureau of Economic Analysis |
| CEDA | Comprehensive Environmental Data Archive |
| CFP | Current Food Consumption Patterns |
| $CO_2$-eq | Carbon Dioxide Equivalent |
| EIA | Energy Information Administration |
| EIO | Environmentally-extended Input-Output |
| ELCD | European Life Cycle Database |
| ERS | Economic Research Service |
| EPA | Environmental Protection Agency |
| FAO | Food and Agriculture Organization |
| FP | Food Patterns |
| GHGEs | Greenhouse Gas Emissions |
| I/O | Input-Output |
| LAFA | Loss-Adjusted Food Availability |
| LCA | Life Cycle Assessment |
| LCI | Life Cycle Inventory |
| MCS | Monte Carlo Simulation |
| MSW | Municipal Solid Waste |
| NAICS | North American Industry Classification System |
| NHTS | National Household Travel Survey |
| RECS | Residential Energy Consumption Survey |
| RFP | Recommended Food Consumption Patterns |
| SFSC | Short Food Supply Chain |
| TRACI | Tool for the Reduction and Assessment of Chemical and other environmental Impacts |
| UK | United Kingdom |
| UN | United Nations |
| US | United States |
| USDA | United States Department of Agriculture |
| WHO | World Health Organization |

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
