# Peer review of "Life Cycle Assessment of Dietary Patterns in the United States: A Full Food Supply Chain Perspective"

_sustainability, doi:10.3390/su12041586_

Round 1

Reviewer 1 Report

Major general comments:

It remains unclear whether the analysis include also the agricultural step of the production/distribution/consumtption entire process. This hall be specified much better, also indicating if the differences accounted for in the comparison include production techniques at this stage. Furthermore, it shall be better specified if different places of production are considered with implications on productivity and coefficients in the use of resources. Implications of all the different hypothesis made in the different scenarios built shall be presented and discussed more clarly. Please specify wether balance sheet at national level are computed and if the trading (import-export) sector is affected in the new scenario as this may have relevant consequences of its environmental impact. It is not clear to me where, in the analysis, do the prices enter and what prices are used and if in the new scenario new market balances are considered; in other words, are prices endogenous? And, if this is the case, how these prices are estimated?

Minor comments:

line 30: …consumers and citizens at large.

Line 89: please check if the expression “truncation errors” is ppropriate in this context.

Lines 121-124 some of the listed impact indicators are overlapped and some do have reciprocal influences. This shall be considered in the analysis.

Figure 1: in the version I got, something seems to be missing, please check-

In the appendix, some of the equivalences have mispecified unit measures (eg: S3 “per year” is lacking, please, check throughout the appendix.

Reviewer 2 Report

Review Report:

Sustainability

Manuscript Number: 698031

Title: Life Cycle Assessment of Dietary Patterns in the United States: A Full Food Supply Chain Perspective

In this paper, the authors used a hybrid input-output based life cycle assessment (LCA) to analyze the potential environmental impacts of the U.S. food supply system. The overall research framework and planning are clear, but the results and analysis still need to be strengthened otherwise it looks like the industrial environment report. There are some comments as below.

As a valuable paper, the Abstract should be considered presenting specific numbers and quantitative results. The Abstract is currently written in a way that has too much data information and too little description of the value and findings, making it impossible to understand the importance of the study. A similar situation occurs in the part of Discussion in Chapter 5. As a final summary, it should not just to compare the results of previous literatures, but should be pointed out specific contributions and clearly different findings. For example, the reasons for differences in important results of American dietary culture. When others may cite your paper in the future, they must know the conditions under which they can correctly cite. The research idea and main contribution should be based on environmental assessment of the overall food supply chain and analysis of different food groups. However, there is a lack of more discussions on the environmental impacts of different stages in the food supply chain, as well as the inference from the food group of strategies that may reduce impacts at each stage of the supply chain. The contribution and analysis of environmental impact reduction in different scenarios should be further clarified and explained. The study provides a complete description of the environmental impacts of the entire US food supply chain, but most focus on data-based results. Uncertainty and sensitivity analysis are important analysis items in LCA, but there is no more accurate and in-depth description of its factor selection and process, and it is recommended that it should be strengthened.

In conclusion, this paper provides a complete description of the information, and the steps of the method have been explained in detail. I think research should be valuable in the field of food environmental impacts, but there is no more in-depth discussion. In the method part, the detail and explanation of the uncertainty and sensitivity analysis should be further explained. Furthermore, it should be based on the case in the United States to explain the reasons and mitigation strategies for others to cite in the future. At present, the presentation still looks like an industrial environment report and cannot be understood as a valuable academic article. However, if the authors can make clear suggestions and explanations for the results, and do more discussion and method construction on the part of uncertainty analysis, I believe that after major revision, it should still be a valuable article.

Reviewer 3 Report

Please use compact captions. Larger information may be reallocated to the body of the text;

At the end of section 2.3, you should include at least one paragraph on SFSC (Short Food Supply Chain) or AFN (alternative food networks). In these kinds of array, the impact of shortened distances in environmental and social concerns are relevant and cannot be ignored in a study that approaches waste in food production. You may wish to refer to https://doi.org/10.1016/j.jclepro.2017.01.118,  and https://doi.org/10.1016/j.jclepro.2017.09.235 to ground your approach;

Please remain at the first indentation level (3.1), avoiding a further level (3.2.1). I don´t think you need three levels;

In section 3.4, please define reverse logistics and reverse channels to route post-consumption waste;

The rest of the article is fine and requires no further concerns.

Round 2

Reviewer 1 Report

Additional comments:

The goals of the paper are presented differently and somehow confusingly in different parts of the text. See lines 78-83, 118-121 and 542, please express them more coherently. Lines396-411: reasons for this unexpected result should be presented and discussed more explicitly and more indepth. Is the effect entirely due to increased quantities? Line 118-121: please align the verbs, either in present or past form. The term “supply” should be replaced by “chain” in specific context where consumption is also meant to be included (e.g. lines 78 and 543), in fact, consumption is not part of supply but it can be considered as part of the whole chain articulation. When using the word “both”, the article “the” should come before (e.g. line 395and 509) Line 510: Furthermore, THE majority of….. (please add the article) Line 511 patterns to A recommended diets. Please remove the article or put to singular (diet) Line533 IT is possible… please, change to: THIS is possible…. Line 548please remove THE. Line563: the subject of the sentence is not clear, please substitute “it” with an explicit subject

Reviewer 2 Report

On the whole, this revision supplements many previously unclear points, and adds many explanations and findings. However, I still think that the management implications of comparing results to situations are quite inadequate. Most of the results are more like a description of the current situation. The uncertainty part has also been reinforced and explained, and overall it looks better than before. However, the author's suggestions under uncertainty are not seen. If the meaning cannot be reinforced, and it is only the case of the United States, I'm afraid it's not very useful to other readers.
